# Gauss Markov and Flow Balanced Vector Radial Learning network traffic classification on IoT with SDN

Rajkumar Kulandaivel[1], Manikandan Ramachandran[1],
Sathishkumar Veerappampalayam Easwaramoorthy[2], Jaehyuk Cho[3]*

1 School of Computing, SASTRA Deemed University, Thanjavur, Tamil Nadu, India, 2 Department of Computing and Information Systems, Sunway University, Petaling Jaya, Selangor Darul Ehsan, Malaysia, 3 Department of Software Engineering, Jeonbuk National University, Jeonju-si, Republic of Korea

* chojh@jbnu.ac.kr

**Data Availability Statement:** All DDOS attack SDN Dataset files are available from the Mendeley data database(URL: https://data.mendeley.com/datasets/jxpfjc64kr/1) (DOI:10.17632/jxpfjc64kr.1).

## Abstract

Recent evolution in connected devices modelled a massive stipulation for network traffic resources and classification. Software-defined networking (SDN) enables ML techniques with the Internet of Things (IoT) to automate network traffic. This helps to reduce accuracy and improves latency. Problems by conventional techniques to categorize network traffic acquired from IoT and assign resources can be resolved through SDN solutions. This manuscript proposes a proposed network traffic classification technique on IoT with SDN called Gauss Markov and Flow-balanced Vector Radial Learning (GM-FVRL). With the network traffic features acquired from the IoT devices, SDN-enabled Gauss Markov Correlation-based IoT Network Traffic Feature Extraction is applied to extort relevant network aspects. Next, the flow-balanced radial-based ML model for network traffic categorization uses the relevant extracted network traffic features. With the aid of flow, the balanced radial basis function reduces the influence of noise due to distinct network flow. This helps to improve accuracy and minimize latency. Due to this, better precision and recall is ensured. Performance of our method has been evaluated utilizing a scheme using an SDN traffic dataset. The results show that our method classifies the network traffic with high classification accuracy and minimum latency, ensuring better precision and recall.

## 1. Introduction

In SDN, IoT has been extensively and widely applied in everyday chores for various applications such as Hospitality, agriculture, etc. The enormous employment and application nature of numerous sensors has resulted in an enormous data volume, causing huge network traffic. Hence, it becomes very notable to ensure the network traffic flows to bestow usual operation of pertinent services of IoT for users. On the other hand, SDN in recent years has emerged as a novel network architecture that distinguishes control as well as broadcast cores of network to control plane as well as data plane. With SDN Traffic Monitoring, network stability is ensured using a timely collection of flow statistics to experiment [1]. Several methods have been proposed in recent years to monitor traffic with Software-Defined Networks.

**Funding:** This work was supported the Korea Environmental Industry & Technology Institute (KEITI), with a grant funded by the Korea government, Ministry of Environment (The development of IoT-based technology for collecting and managing big data on environmental hazards and health effects), under Grant RE202101551.

**Competing interests:** The authors declare that they have no known competing financial interests or personal relationships that could have appeared to influence the work reported in this paper.

A method called IPro was proposed in [1] to address the tradeoff between probing and accuracy of network traffic. Knowledge-defined Networking, which bestowed an effective resolution involved in traffic analysis, was initially designed. Fine-tuning follows this aim to retain the CCO at the threshold level. Next, a method depending on the design of R-L was introduced that measured the concerned probing interval considering the traffic variations like CCO and CUC.

Through evolution of IoT, dissimilar QoS of numerous information flows of different IoT services have to be guaranteed.SDN is contemplated as a novel model that ensures the QoS necessitate different services by making the efficient separation between control logic. However, intelligent mechanisms are not required for detecting trade-offs with learning as network behaviour. To solve this issue, a network traffic categorization technique on IoT through SDN called GM-FVRL is designed to extract the computationally efficient and relevant network traffic feature for classification from SDN. MACCA2-RF&RFis, termed as Random Forest, was proposed in [2] using a novel data flow classification using machine learning methods. This ensured accuracy and minimized the time consumed in QoS necessities for information flows. However, knowledge networking cannot be directly utilized for optimizing network traffic classification because of its vast demand for flow tables and the compromise of the latency rate. QoS offers a network for particular data broadcast in an assured feature.

Additionally, the intelligent routing method only works in different data flows that do not embrace the network through dynamic alters. The network traffic classification method employs ML to attain comprehensive and precise optimization in uninterrupted time, which is a large ultimatum. However, generating enormous amounts of IoT information of dissimilar services makes it more complex for conventional networks to ensure QoS necessities of information flow. To solve this issue, GM-FVRL is designed.

To address the challenges, we propose the GM-FVRL method to minimize the latency in extracting relevant network traffic features in the SDN with the flow-balanced radial-based machine learning model. The contributions of GM-FVRL are listed as follows.

- First, GM-FVRL optimizes customized Gauss Markov worthiness function and obtains computationally efficient and optimal network traffic features by intelligently adjusting the F1-measure function.

- Second, GM-FVRL architecture saves round trip time and enhances accuracy for separating data plane and control plane via SDN from the corresponding IoT devices.

- Third, the flow-balanced radial-based machine learning method can efficiently attain network traffic classification's precision and recall rate by employing the flow-balanced radial basis function.

- We estimate the performance of GM-FVRL through experiments. Experimental outcomes demonstrate that GM-FVRL has superior latency and attains enhanced performance and stability in classification accuracy and precision compared to conventional network traffic classification methods.

The rest of this manuscript is structured as follows. In section 2, literature surveys are explained. The network traffic classification method process on IoT with SDN is called GM-FVRL, and the establishment of the SDN-enabled IoT Network method and the procedure of network traffic flow classification are explained in section 3. Section 4 is a detailed study of results, and the manuscript is summarized in section 5.

## 2. Literature survey

An evolving network architecture using SDN was presented in [3], and an elaborative review was also addressed, providing information regarding components like design, programmability, etc. An efficient load-balancing method employing a genetic programming model, GPLB, was proposed [4], contributing to latency and overhead. However, effective network traffic flow classification in the prevailing networking domain is laborious and restrictive. To overcome this problem, GM-FVRL is developed. A new lightweight network traffic flow categorization technique for SDN applications was proposed in [5] using flow-based forwarding to extract traffic statistics from the data plane.

A review of DL-based traffic categorization at SDN was investigated in [6]. DROM method for SDN to attain holistic routing was proposed in [7]. However, the designed method minimizes latency and enhances throughput. To overcome this problem, GM-FVRL is developed. With the aid of DROM, network operation was simplified, and network performance maintenance was enhanced, including delay, throughput, time, and so on. In [8], SDN architecture and RBFNN were intelligently integrated for network traffic forecast. However, with a high dimensionality nature, network traffic classification is in the preliminary stage.

Deep learning was applied in [9], where the network traffic features were extracted based on auto-encoder, extracting representative network features to represent high dimensional data. The complication involved in analyzing the new types of attacks is said to be a critical issue. In [10], a novel hybrid DL model based on a CNN was proposed to categorize network traffic flow. Here, a novel regularizer technique has been employed to mention overfitting issues and enhance the potentiality of intrusion. A network slice based on machine learning was proposed in [11] to address the data convergence.

A logistic regression method was proposed in [12] to improve the true positive ratio involving network traffic classification. Nevertheless, another bandwidth control mechanism in SDN against attacks was presented in [13], improving the detection accuracy significantly. However, a significant detection delay was observed, compromising the overall classification rate. To address this issue, ensemble learning in SDN was proposed in [14] that identifying the typical characteristics of network traffic ensured timely delivery.

A survey of IoT management based on SDN was investigated in depth [15]. A machine learning algorithm was applied in [16] for efficient detection and classification of network traffic flows. Nevertheless, another online traffic categorization model that depends on DL was presented [17], focusing on accuracy and precision. A DT and SVM were applied for fast detection and classification [18], achieving accuracy. Euclidean Distance-based Multiscale Fuzzy Entropy was designed to detect numerous network traffic anomalies [19]. However, still certain drawbacks to conventional methods. The existing methods concentrated on the precision and accuracy aspects. However, the latency and classification accuracy of network traffic flow were not focused on. In our review of recent advancements in software-defined networking (SDN), we conducted a comparative analysis of intelligent methods proposed by various researchers (see Table 1). In the forthcoming sections, we propose a novel GM-FVRL technique to mention the shortcomings of existing techniques.

## 3. Methodology

Conventional switches modernize routing tables and include data flow information of the destination node. However, the SDN controller has a multifaceted outlook of the network. Through modernizing topology data, SDN controller discover every relevant network feature while analyzing the network traffic from the data plane on IoT. The network design for GM-FVRL method is depicted in Fig 1. To begin through network traffic traces provided as

**Table 1. Comparative analysis of intelligent methods in software-defined networking (SDN) research.**

| SI. No | Author (year) | Title | Method | Proposal Contribution | Merits | Demerits |
|---|---|---|---|---|---|---|
| 1. | Edwin F. Castillo, Lisandro Z. Granville, Armando Ordonez, Oscar Mauricio Caicedo Rendon. (2020), Computer Networks, Elsevier. | IPro: An Approach for Intelligent SDN Monitoring | IPro method | A method called IPro was proposed in [1] to address the tradeoff between probing and the accuracy of network traffic. Knowledge Defined Networking (KDN-based architecture) to an effective solution for fine-tuning the probing interval involved in traffic analysis with SDN was initially designed | IPro ensures QoS necessitate different services for efficient separation among control logic from data plane | However, intelligent mechanisms are not required for detecting trade-offs with learning as network behaviour |
| 2. | Weifeng Sun, Zun Wang, Guanghao Zhang. (2020) Computer Networks, Elsevier, Nov 2020 [MACCA2-RF&RF] | A QoS-guaranteed intelligent routing mechanism in software-defined networks | Intelligent routing MACCA2-RF&RF method | An intelligent routing method called MACCA2-RF&RF was proposed in [2] using a novel data flow classification using machine learning methods. | Time consumption is reduced in QoS requirements for data flows. | Knowledge Defined Networking is not applied for optimizing network traffic classification as there is a huge demand for flow tables and compromising latency rate. The generation of massive IoT data from different services makes it more and more difficult for traditional networks to guarantee the QoS requirements of data flow. |
| 3. | Nitheesh Murugan Kaliyamurthy, Swapnesh Taster, Suresh Shanmugasundaram, Ankit Saxena, Omar Cheikhrouhou, Hadda Ben Elhadj. (2021) | Software-Defined Networking: An Evolving NetworkArchitecture—Programmability and Security Perspective | Network architecture using SDN | An evolving network architecture using SDN was presented in [3] which an elaborative review was also addressed, providing regards to components like design, programmability, and so on | Networking domain attempts to propose cost-efficient, effective and feasible solutions. | It has a limited perspective. |
| 4. | Shahram Jamali, Amin Badirzadeh, Mina Soltani Siapoush. (2019) Digital Communications and Networks, Elsevier | On the use of genetic programming for balanced load distribution in software-defined networks | Efficient load balancing method | An efficient load-balancing method employing a genetic programming model, called Genetic Programming Load Balancing (GPLB), was proposed in [4] for contributing latency and overhead | Latency and jitter are minimized. | However, effective network traffic flow classification in the prevailing networking domain is laborious and restrictive. |
| 5. | Yuqing Wang, Dingde Jiang, Liuwei Huo, Yong Zhao. (2019) Mobile Networks and Applications, Springer. | A New Traffic Prediction Algorithm to Software-Defined Networking | Lightweight network traffic flow classification method | A novel lightweight network traffic flow classification method for SDN applications was proposed in [5] using flow-based forwarding to extract traffic statistics from the data plane. | The lightweight network traffic flow classification method is feasible and effective. | Effectively predicting network traffic in current networks is very difficult and nearly prohibitive. |

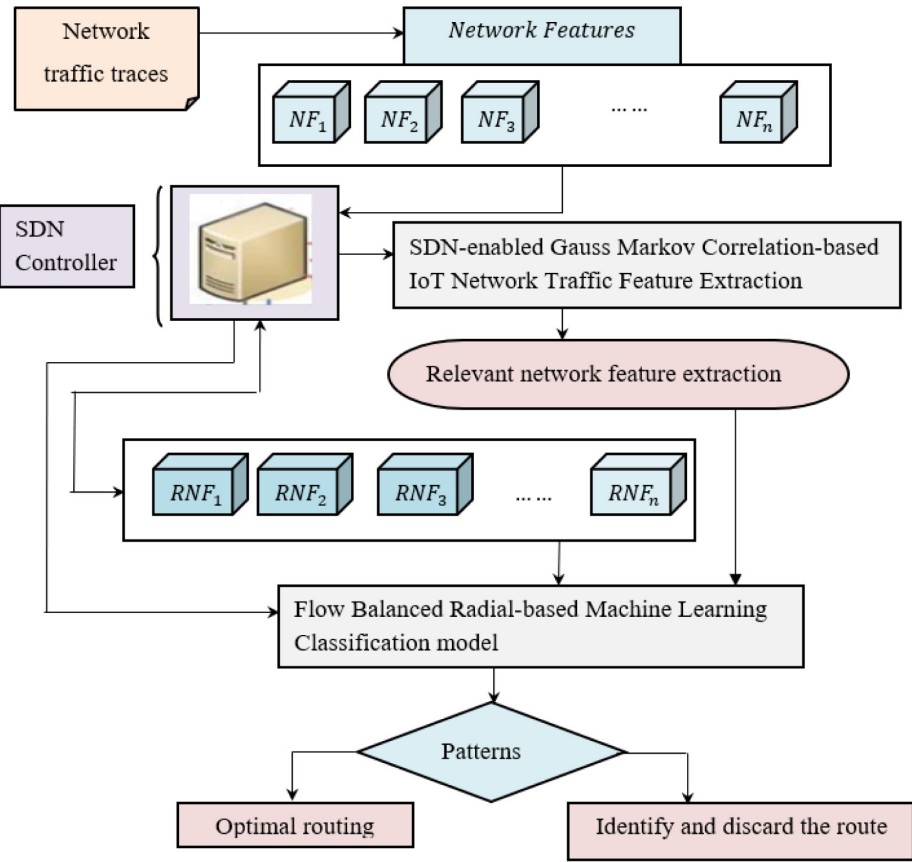

**Fig 1. Structure of the Gauss Markov and Flow-balanced Vector Radial Learning (GM-FVRL) method.**

input obtained from the devices, the SDN controller applies a Gauss Markov Optimal Mean Correlation-based Network Traffic Feature Extraction model to extract the relevant network feature. Then, with the obtained relevant network feature, the SDN controller applied a classification model to categorize network traffic as benign or malicious, and accordingly, either optimal is ensured by following network traffic through benign class and discarding route in case of malicious. Fig 1 shows the structure of the GM-FVRL method.

The above figure shows that the GM-FVRL method is split into three sections. First, the raw network traffic traces are obtained from the SDN traffic dataset [20]. Next, the relevant network traffic features are extracted using the SDN-enabled Gauss Markov Correlation-based IoT Network Traffic Feature Extraction method. After that, with computationally adequate and relevant aspects, the network traffic classification is performed using the flow-balanced radial-based machine learning classification model. With the obtained results, i.e., the normal patterns, either optimal routing is ensured or the route is discarded in case of abnormalities or malicious functioning. An elaborate description of the proposed method, followed by the network system model, is provided in the following subsections.

### 3.1. SDN-enabled IoT Network model

Unlike the conventional traffic classification method that only depends on the controller, a novel network traffic categorization technique depends on SDN. The holistic network traffic

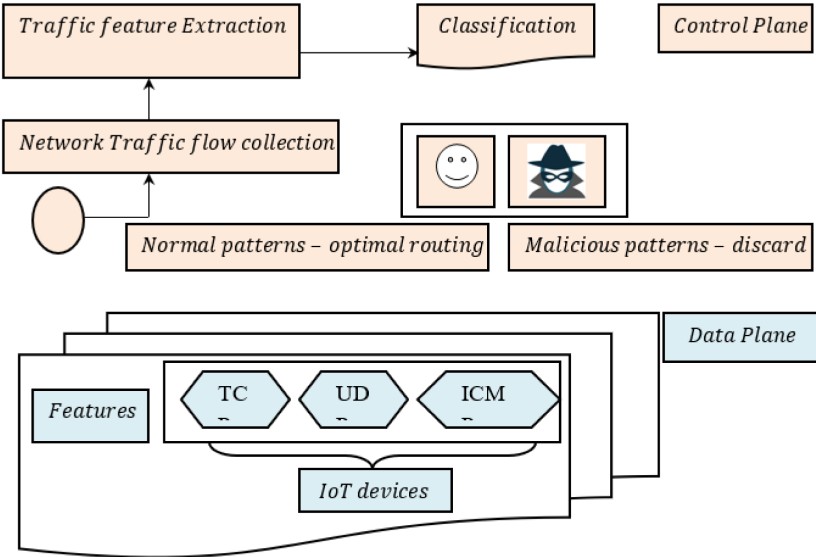

**Fig 2. Architecture of the network traffic classification method on IoT with SDN.**

classification is split into data and control planes. The main structure of the SDN-enabled IoT Network model is shown in Fig 2.

Fig 2 illustrates the Network traffic classification method on IoT with SDN. First, we set up the preliminary network traffic feature extraction model in edge switch in the data plane. The Lightweight Gauss Markov Correlation-based Network Traffic Feature Extraction method depends on correlation. It is used to extract relevant network traffic features and report to the controller about the same. In the meantime, the controller at control pane leads every switch of the corresponding size of the data packet along with the duration. With this, either the optimal route is ensured or the route is said to be discarded. According to the network traffic classified results using Flow Balanced Radial-based ML method for network traffic classification algorithm.

IoT networking has many resource-constrained devices which have recently become in demand. Nowadays, IoT systems rely mainly on TCP/IP protocols. In IoT, UDP was less than TCP. However, UDP provides IoT, which employs fewer network resources for transferring and does not protect the connection between two endpoints. ICMP is the protocol device in the network used for communication issues by data transmission. ICMP is employed to obtain and receive information at the destination at the correct time.

As shown in above figure, the software-defined network is an improvement of custom network visualization. This technology divides control plane management of network devices as of underlying data plane which forwards network traffic. Bottom layer of the SDN constitutes data plane. The data plane refers to data packet forwarding devices, routers, and access points. SDN topology is modelled using graph '$G = (V,E)$', where '$V = V_1, V_2, V_3,\ldots,V_n$' here refers to the set of switches, and '$E = E_1, E_2, E_3,\ldots,E_n$' corresponds to the links that connect the switches. Also, every switch is associated with the controller that acquires network traffic data [20] in a presumed time interval. The network traffic information includes 23 features, both extracted

and calculated. The network traffic information matrix is shown in Eq 1.

$$NTIM = \begin{pmatrix} NT_{E1,1} & NT_{E1,2} & NT_{E1,3} & \ldots & NT_{E1,n} \\ NT_{E2,1} & NT_{E2,2} & NT_{E2,3} & \ldots & NT_{E2,n} \\ NT_{E3,1} & NT_{E3,2} & NT_{E3,3} & \ldots & NT_{E3,n} \\ \ldots & \ldots & \ldots & \ldots & \ldots \\ NT_{Em,1} & NT_{Em,2} & NT_{Em,3} & \ldots & NT_{Em,n} \end{pmatrix} \tag{1}$$

The elements of network traffic information matrix '$NTIM$' that is referred to as. '$NT_{Ei,j}$' possess someone of three distinct values. First case '$NT_{Ei,j} = -1$', refers to the state that there is no association among switches '$i$' and '$j$', and no network traffic flow subsists between the two devices. On the other hand, the second case '$NT_{Ei,j} = 0$' refers to the state that there is an association among switches '$i$' and '$j$'. Network traffic flow subsists between two devices following a connection. Finally, the third case '$NT_{Ei,j} \geq 0$', refers to the state that there is a direct association between switches '$i$' and '$j$', network traffic flow subsist. Finally, the value denotes the sum of data packets conceded on the link among switches '$i$' and '$j$'.

## 3.2. SDN-enabled Gauss Markov Correlation-based IoT Network Traffic Feature Extraction

One of the significant challenges in Software Defined Networks is the target category traffic identification for network data traffic classification. Network traffic incorporates network behaviour patterns and user activity patterns that bear numerous intrinsic and essential network traffic features and dynamic characteristics acquired from the IoT devices. On the other hand, the SDN's core idea comprises a decoupled control plane and data plane. The network condition is rationally centralized, and the controller is abstracted as of underlying network ability. Also, in SDN, due to logically centralized nature of the controller, enormous traffic is said to be generated and stored instantaneously in the corresponding IoT devices. With the growing network traffic information, ML models are dreadful and act significantly in acquiring knowledge of stored network traffic data and further classification. Extracting network traffic features, as with the manual selection model, is laborious and cumbersome to extract the network traffic features accurately. This work proposes a Gauss Markov Optimal Mean Correlation-based Network Traffic Feature Extraction model in SDN. Fig 3 shows the

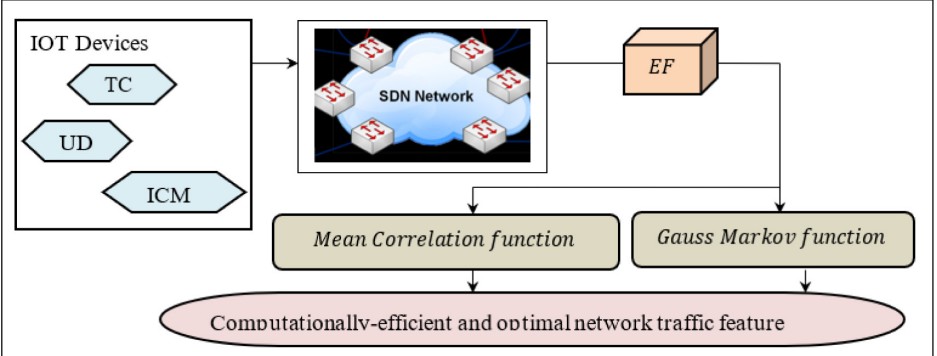

**Fig 3. Structure of Gauss Markov Correlation-based Feature Extraction model.**

structure of the SDN-enabled Gauss Markov Correlation-based IoT Network Traffic Feature Extraction model.

In Fig 3, GMC-FE extracts the network feature subset '*FE*' and hence can be utilized to classify two class labels of the SDN dataset [20]. The GMC-FE algorithm identifies the network subset of features to minimize the dataset dimensionality and enhance the classification accuracy. The GMC-FE defines worthiness for each selected subset of network features. The worthiness is estimated based on the Gauss-Markov proposition that an optimistic subset of network features has arrived. It is less correlated to every other, but it is significantly correlated to the class label, also with the lowest sampling variance within the class) label. The Gauss Markov Worthiness is mathematically defined as a formula(Eq 2).

$$GMW = \sum \beta_j X_{ij} \left[ \frac{n * Cor[M_{FC}]}{\sqrt{n + n(n-1)Cor[A_{FF}]}} \right] \ Where \ i, j = 1, 2, 3, \dots, n \qquad (2)$$

From the above Eq (2), the Gauss Markov worthiness '*GMW*' is estimated based on the network feature subset containing '*n*' network features, mean network feature class correlation '*Cor*[$M_{FC}$]' and the average feature class correlation '*Cor*[$A_{FF}$]' respectively. In addition, unobservable network traffic features '$\beta_j$' and observable network traffic feature '$X_{ij}$' are included to arrive at the postulate. The observable network traffic feature refers to the extracted features [in Table 2], and the unobservable network traffic feature refers to the

**Table 2. Dataset description.**

| S. No | Features | Description |
|---|---|---|
| 1. | Protocol | TCP, UDP, ICMP |
| 2. | Extracted features | 1. Switch—Switch-id |
| | | **2. Packet_count–(pktcount)** |
| | | 3. byte_count–(byte count) |
| | | **4. duration_sec, (dur)** |
| | | 5. duration_nsec, which is the duration in nanoseconds (dur_nsec) |
| | | 6. total duration is the sum of duration_sec and durstaion_nsec, (tot_dur) |
| | | 7. Source IP, (src) |
| | | 8. Destination IP, (DST) |
| | | **9. Port number, (port_no)** |
| | | **10. tx_bytes is the number of bytes transferred from the switch port (tx_bytes)** |
| | | 11. rx_bytes is the number of bytes received on the switch port (rx_bytes) |
| | | 12. dt field shows the date and time, which has been converted into the number (dt) |
| | | **13. Flow is monitored at a monitoring interval of 30 seconds. (flows)** |
| 3. | Calculated features | 1. Packet per-flow, which is packet count during a single flow (pktperflow) |
| | | 2. Byte per flow is the byte count during a single flow, (byteperflow) |
| | | 3. Packet Rate is the number of packets sent per second and calculated by dividing the packet per flow by monitoring interval (pktrate) |
| | | 4. number of Packet_ins messages, (packetins) |
| | | 5. total flow entries in the switch (pair flow) |
| | | 6. tx_kbps, (tx_hbps) |
| | | 7. rx_kbps are data transfer and receiving rate, and (rx_kbps) |
| | | 8. Port Bandwidth is the sum of tx_kbps and rx_kbps |
| 4 | Class label | 1. Benign (0) or Malicious (1) |

calculated features [in Table 1]. For each network feature subset, the mean network feature class correlation '$Cor\,[M_{FC}]$' is mathematically formulated in Eq (3).

$$Cor\,[M_{FC}] = \frac{\sum_{i=1}^{n} F_i}{n} \tag{3}$$

From the above Eq (3), mean network feature class correlation is estimated based on the average of the network features '$F_i$' respectively. Similarly, the average feature class correlation '$Cor\,[A_{FF}]$' is mathematically formulated in Eq (4).

$$Cor\,[A_{FF}] = \frac{F_1 + F_2 + F_3 + \cdots + F_n}{n} \tag{4}$$

From the above Eq (4), average feature class correlation is evaluated based on the sum of all the network features '$F_1 + F_2 + F_3 + \cdots + Fn$' and the total number of network features '$n$', respectively. However, there are certain network traffic features with redundant information. Therefore, the '$n$' network feature subset is processed to a smaller and more optimal extraction. Then, the network traffic features are extracted based on the F1 measure employing precision and recall as in Eq (5).

$$F1\,[F(n)] = \frac{2*Precision\,[F(n)]*Recall\,[F(n)]}{Precision\,[F(n)] + Recall\,[F(n)]} \tag{5}$$

The Network Traffic Feature Extraction model captures network traffic and extracts the most relevant network traffic features along with network environment changes. Finally, the optimal network traffic feature is the feature that carries out the optimality on F1-measure simultaneously with the already extracted features. This is mathematically formulated in Eq (6):.

$$E = argmax\left\{ F1\left[\bigcup_{i=1}^{FE-1} F_i \bigcup F(n)\right]\right\} \tag{6}$$

Finally, the optimal network traffic features extracted '$E$' are evolved based on the aggregated results of the '$F1$' and the already extracted feature '$\bigcup_{i=1}^{FE-1} F_i \bigcup F(n)$'.

**Algorithm 1. Gauss Markov Correlation-based Network Traffic Feature Extraction**.

**Input:** Dataset '$DS$', Features '$F = F_1,\ F_2,\ F_3,\ \ldots, F_n$'
**Output:** Computationally efficient and optimal network traffic feature extraction '$E = E_1,\ E_2,\ E_3,\ \ldots, E_n$'
1: **Initialize** '$n = 23\ features$', mean network feature class correlation '$Cor\,[M_{FC}]$', average feature class correlation '$Cor\,[A_{FF}]$'
2: **Begin**
3: **For** each Dataset '$DS$' with network feature subset consisting of '$n$' features
4: Estimate the worthiness '$W$' as in Eq (2)
5: Estimate network traffic features extracted based on the F1-measure as in Eq (5)
6: Obtain optimal network traffic features as in Eq (6)
7: **Return** (optimal network traffic features)
8: **End for**
9: **End**

As given in the above Gauss Markov Correlation-based Network Traffic Feature Extraction algorithm, to reduce the latency and improve the network traffic classification inaccuracy. An integrated Gauss Markov function and Optimal Correlation-based function are employed. First, with the aid of the Gauss Markov function, minimum variance involved in optimal

network traffic feature extraction is ensured, contributing to better accuracy. Also, a relevant network traffic feature for classification is obtained by extracting the Optimal Correlation-based function. Therefore reducing the latency involved in the overall process.

## 3.3. Flow balanced radial-based ML method for network traffic categorization

The controller, the SDN operating scheme, is in charge of classifying networks and sustaining numerous processes. Despite all its potentialities, the development of numerous architectural structures constitutes several threats. This section proposes the flow-balanced radial-based machine learning (FBR-ML) model for network traffic categorization to categorize network traffic features on IoT with SDN.

An enhanced kernel function, FB-RBF, is introduced to minimize noise through network traffic flow dissimilarity. It develops an FBR-ML method which forecasts whether a novel sample or the network traffic flow falls into one of the classes. In other words, upon exceeding the threshold value for the count of a data packet, the class is considered malicious, and the identified route is discarded. On the other hand, the class is considered normal, therefore ensuring optimal routing. Let us consider the sample data (i.e., taking into consideration the feature extracted) '$S = \{(E_1, y_1), (E_2, y_2), \ldots, (E_n, y_n)\}$', where '$E_i \in R^n$','$y \in \{+1, -1\}$'. The '$E_i$' denotes the extracted feature input vector, and '$y$' represents the resultant value. Here, the network traffic classified output contains two values, '+1' or '−1'. As for extracted feature inputs, an optimal hyper-plane is drawn, which divides the information into distinct classes and is expressed below.

$$f(E) = Sign\,(W, E) + b \qquad (7)$$

From the above Eq (7), considering the resultant weight values '$W$' for the respected extracted features '$E$' along with the bias value '$b$', the signed outputs provide the result either of the two classes. Next, an enhanced kernel function called FB-RBF is designed to reduce noise due to differences in network traffic flow.

$$K\left(E_i,\, E_j\right) = EXP\left[\dfrac{-\left(\frac{E_i - M}{SD} - \frac{E_j - M}{SD}\right)^2}{\sigma^2}\right] \qquad (8)$$

From the above Eq (8), '$E$, $M$, $SD$' denotes the sample extracted network feature vector, mean of the extracted network feature vector. The standard deviation of the extracted network feature vectors, respectively. Moreover, '$M_i, SD_i$' is mathematically expressed as given below.

$$M_i = \frac{1}{n}\sum\nolimits_{j=1}^{n} E_{ij} \qquad (9)$$

$$SD_i = \sqrt{\frac{1}{(n-1)}\sum\nolimits_{j=1}^{n}\left(E_{ij}\right)^2} \qquad (10)$$

From the above Eqs (9) and (10), '$n$' denotes the training samples and '$E_{ij}$' corresponds to

the '$i - th$' attribute of the '$j - th$' sample.

$$W(E) = \begin{cases} -1, \; if \left( tx_{bytes} < 3500 \right) \; and \; (dur < 15) \; and \; (pktcount < 5000), \\ \qquad\qquad then \; malicious, \; discard \; route \\ \qquad +1, \; then \; normal, \; optimal \; routing \end{cases} \qquad (11)$$

With the above resultant weight values at the controller of the SDN, network traffic flow is classified as usual, following optimal routing. Alternatively, network traffic flow is categorized as malicious after discarding the route.

Algorithm 2. Flow balanced radial-based machine learning model for network traffic classification.

```
Input: Dataset 'DS', Features 'F = F₁, F₂, F₃, ....,Fₙ'
Output: Precise network traffic classification
1: Initialize feature extracted 'E = E₁, E₂, E₃, ...,Eₙ'
2: Begin
3: For each Dataset 'DS' with network feature subset consisting of 'n'
   features
4: Estimate optimal hyper-plane as in Eq (7)
5: Evaluate Flow Balanced Radial Basis Function as in Eq (8)
6: If 'W(E) = −1',
7: Then malicious route and discard route
8: End if
9: If 'W(E) = +1',
10: Then normal route and perform routing
11: End if
12: End for
13: End
```

As given in the above Flow Balanced Radial-based ML model for network traffic categorization algorithm, for performing network traffic categorization with high precision and recall. With this objective, an SVM model is designed to address the noise occurring due to distinct traffic flow. A flow-balanced radial basis function is employed to contribute to higher precision. Next, with the enhanced kernel function, the factors or features contributing to the network traffic analysis are modelled in the weight, ensuring high recall.

## 4. Experiment setup

We converse a comparative revision of Gauss Markov and Flow-balanced Vector Radial Learning (GM-FVRL) with existing methods IPro [1], MACCA2-RF&RF[2], and new hybrid DL method depending on CNN [10] in different metrics. For experimentation, the following data set and simulation environment are examined.

### 4.1. Dataset assortment

For training and testing purposes, the contemporary SDN traffic dataset, https://data.mendeley.com/datasets/jxpfjc64kr/1 [20], is considered and simulated in Python. It is an SDN-specific data set and is employed for network traffic classification. Network simulation runs for benign TCP, UDP, and ICMP malicious traffic. It consists of 23 features, among which specific features are extracted from switches, and others are computed.

The above-explained network traffic classification methods were performed on a machine containing a core i5 processor, 8 GBRAM, 64-bit OS, clock speed of 2.30 GHz.

## 4.2. Experimental analysis

Performance assessment of GM-FVRL and conventional IPro [1], MACCA2-RF&RF [2], and Novel hybrid DL model based on CNN [10] are compared with specific parameters for distinct numbers of network traffic samples.

**4.2.1. Case 1: Latency.** It is estimated by sending a data packet delivered to the sender, where round-trip time is called latency. It is said to be as little as probable. On the other hand, the extreme latency results in bottlenecks. Therefore, it prevents information packets from being sent via links and minimises the links' bandwidth. This is estimated as given below.

$$L = \sum_{i=1}^{n} S_i * Round\_Trip\_Time \left[ port\_no_i\_port\_no_j \right] \tag{12}$$

From Eq (12), latency '$L$' is calculated depending on network traffic samples. '$S_i$'. The round trip time between two different ports '$port\_no_i$' and '$port\_no_j$'. It is calculated in milliseconds (ms). Latency comparison of proposed GM-FVRL versus existing IPro [1], MACCA2-RF&RF [2], and Novel hybrid DL model based on CNN [10] is depicted in Table 3.

Table 2 given above illustrates the latency for 5000 distinct network traffic samples. The comparison of $L$ is examined during dataset through three conventional methods. The table illustrates a rising trend with increased network traffic samples. This increases the bytes being transferred from the switch port, causing latency. From comparison analysis, it is inferred that the latency is comparatively better when applied using the GM-FVR, while simulations are conducted with 500 network traffic samples. It was observed that 517.5ms were used for GM-FVRL, 557.5ms were used for [1], 592.5ms were used for [2], and 610.55ms were used for [10]. From these outcomes, it is surmised that the latency using the proposed GM-FVR is found to be better than existing [1, 2, 10]. The latency improvement was the incorporation of Gauss Markov Correlation-based Feature Extraction, which is used in the proposed GM-FVR. The relevant and computationally efficient features were extracted, and further processing was performed with these extracted features. With this, the round trip time was found to be minimized using GM-FVR by 16%, 31%, and 36% compared to [1, 2, 10].

**4.2.2. Case 2: Classification accuracy.** Second metric of importance is classification accuracy. The higher the classification of network traffic samples, the more influential the technique is. It is expressed as below.

$$CA = \sum_{i=1}^{n} \frac{S_{CC}}{S_i} * 100 \tag{13}$$

**Table 3. Latency comparison of proposed GM-FVRL versus existing IPro [1], MACCA2-RF&RF [2], and novel hybrid DL model based on CNN [10].**

| Network traffic samples | Latency (ms) | | | |
|:---:|:---:|:---:|:---:|:---:|
| | GM-FVRL | IPro | MACCA2-RF&RF | Novel hybrid DL model based on CNN |
| 500 | 517.5 | 557.5 | 592.5 | 610.55 |
| 100 | 585.35 | 735.05 | 900.25 | 1025.25 |
| 1500 | 655.15 | 895.55 | 1035.55 | 1150.45 |
| 2000 | 735.25 | 935.35 | 1125.55 | 1255.15 |
| 2500 | 900.15 | 1055.35 | 1325.55 | 1465.75 |
| 3000 | 1035.25 | 1135.55 | 1515.35 | 1650.25 |
| 3500 | 1155.35 | 1315.55 | 1725.55 | 1840.65 |
| 4000 | 1285.45 | 1425.55 | 1955.35 | 2010.55 |
| 4500 | 1325.55 | 1615.35 | 2025.55 | 2175.75 |
| 5000 | 1555.15 | 1925.55 | 2135.55 | 2290.55 |

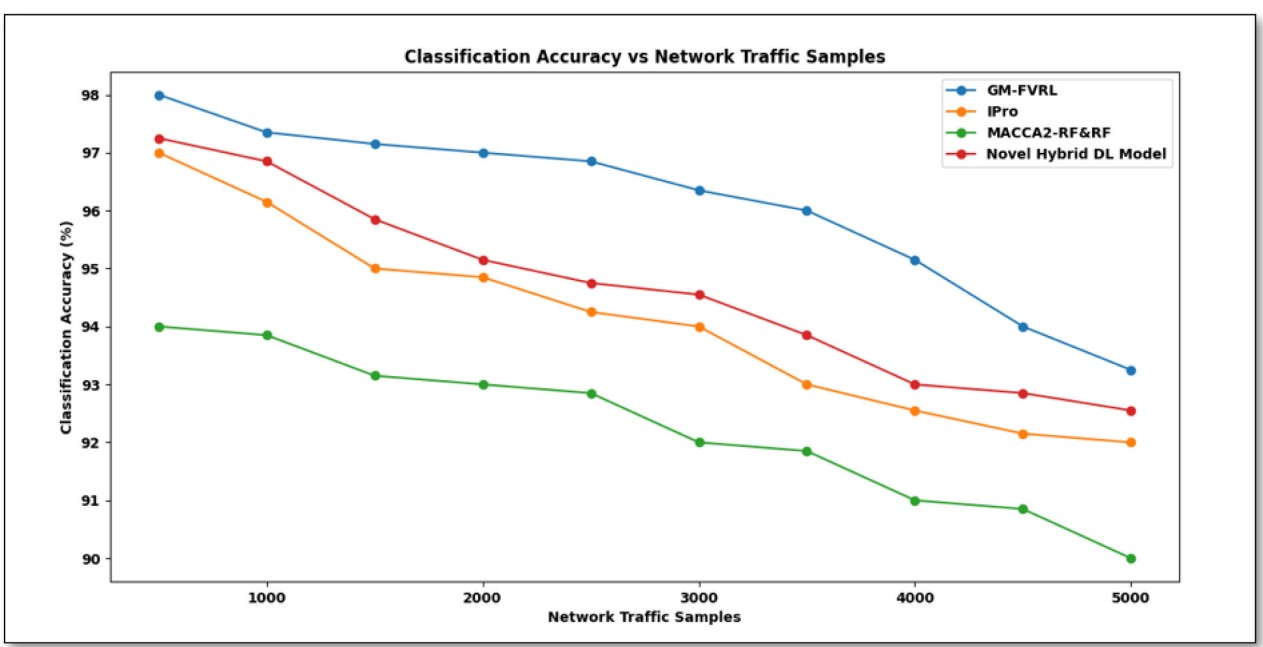

**Fig 4. Classification accuracy comparison of proposed GM-FVRL versus existing IPro [1], MACCA2-RF&RF [2], and novel hybrid DL model based on CNN [10].**

From Eq (13), classification accuracy '*CA*' is calculated depending on the samples '$S_i$' and samples which are correctly classified. '$S_{CC}$'. It is measured in percentage (%).*CA* Comparison of proposed GM-FVRL versus existing IPro [1], MACCA2-RF&RF [2], and Novel hybrid DL model based on CNN [10] is shown in Fig 4.

Fig 4 above shows the classification accuracy for 5000 network traffic samples. The figure x-axis represents the network traffic samples obtained from 500 to 5000 at different time instances and sources for varied durations. It is inferred from the figure that *CA* is inversely comparative to network traffic samples or increasing the network traffic samples causes a reduction in *CA*. However, simulations carried out through 500 samples showed that 490 samples were correctly classified using GM-FVRL, 485 samples using [1], 470 samples using [2], and 486 samples using [10]. This outcome *CA* was found to improve using the proposed GM-FVRL upon comparison with exiting [1, 2]. The higher classification accuracy is due to the Gauss Markov Correlation-based Network Traffic Feature Extraction algorithm proposed by GM-FVRL. The relevant features were extracted using the Gauss Markov function. With these relevant features, classification was made, therefore improving the accuracy rate using GM-FVRL was improved by 2%,4% and 2% than the the [1, 2, 10].

**4.2.3. Case 3: Precision.** It is defined as percentage ratio of pertinent classified events among the retrieved events. This is computed as below.

$$P = \frac{TP}{TP + FP} * 100 \tag{14}$$

From Eq (14), precision rate '*P*' is computed depending on the true positive rate '*TP*' and false positive rate '*FP*'. It is calculated in percentage (%). The precision comparison of proposed GM-FVRL versus existing IPro [1], MACCA2-RF&RF [2], and the Novel hybrid DL model based on CNN [10] is demonstrated in Table 4.

**Table 4. Precision comparison of proposed GM-FVRL versus existing IPro [1], MACCA2-RF&RF [2], and novel hybrid DL model based on CNN [10].**

| Network traffic samples | Precision (%) | | | |
|---|---|---|---|---|
| | GM-FVRL | IPro | MACCA2-RF&RF | Novel hybrid DL model based on CNN |
| 500 | 96 | 94 | 90 | 94.55 |
| 1000 | 95.25 | 93.15 | 89.55 | 94 |
| 1500 | 94 | 92 | 89 | 92.55 |
| 2000 | 94.05 | 91.85 | 90 | 92 |
| 2500 | 94.55 | 93 | 90.55 | 93.25 |
| 3000 | 95 | 93.25 | 92 | 93.75 |
| 3500 | 95.25 | 94 | 92.55 | 94.35 |
| 4000 | 94.15 | 92.15 | 91.35 | 92.65 |
| 4500 | 94 | 92 | 90 | 92.55 |
| 5000 | 94.85 | 93 | 91 | 93.45 |

Table 3 displays classification precision designed in this work, GM-FVRL and [1, 2, 10]. Precision, also called positive prognostic value, corresponds to the percentage ratio of several accurate observations (i.e., correctly classified events) of positive samples and the whole number of each positive observation. The table compared the influence of ML classification method utilized in present study and those used in [1, 2, 10] in terms of precision. From comparison analysis, [1, 2, 10] achieved 94%, 90%, and 94.55% precision, respectively. In this study, a simulation of 500 network traffic samples, they achieved 96% precision using the proposed GM-FVRL. This is due to applying the flow-balanced radial-based ML method for the network traffic categorization method introduced. In this paper, it is found to be comparatively better than that reported in [1, 2, 10] to perform relevant feature extraction steps. The flow-balanced radial-based ML method for the network traffic categorization algorithm is presented. This study assisted in ensuring better precision values of those in [1, 2, 10]. With this, the overall precision was enhanced using GM-FVRL by 2%,5%, and 2% compared to the [1, 2, 10].

**4.2.4. Case 4: Recall.** Recall refers to percentage ratio of relevant classified events which were retrieved.

$$R = \frac{TP}{TP + FN} * 100 \tag{15}$$

From Eq (15), recall 'R' is calculated depending on 'TP' representing correctly predicted network traffic, while the false negative rate 'FN' denotes the misclassified network traffic. It is calculated in percentage (%). Recall comparison of proposed GM-FVRL versus existing IPro [1], MACCA2-RF&RF [2], and Novel hybrid DL model based on CNN [10] is shown in Fig 5.

Fig 5 given above gives R values of network traffic classifiers attained in work compared with [1, 2, 10]. From comparison analysis, it is inferred that the recall values are comparatively better when applied using the GM-FVRL method, while simulations are conducted with 500 network traffic samples. It was observed to be 96.96% using GM-FVRL method, 94.94% using [1], 93.75% using [2], and 95.25% using [10], respectively. From these outcomes, it was inferred that recall by the GM-FVRL technique is found to be better than [1, 2, 10]. This is due to the GM-FVRL using an enhanced kernel function called FB-RBF applied to the learning model. R Parameters enhancement was attributed to relevant network traffic feature extraction steps performed on the dataset. This, in turn, aided in the improvement of R using GM-FVRL by 3%,5%, 2% than the [1, 2, 10] respectively.

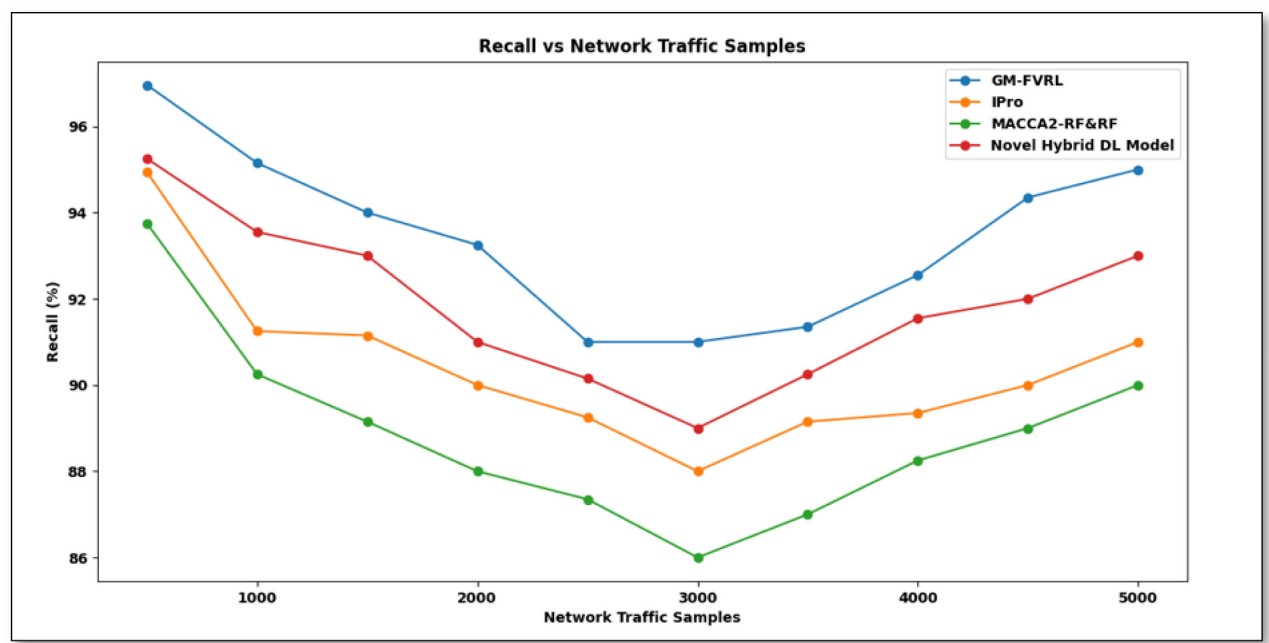

**Fig 5. Recall comparison of proposed GM-FVRL versus existing IPro [1], MACCA2-RF&RF [2], and novel hybrid DL model based on CNN [10].**

**4.2.5. Case 5: Error rate.** It is the ratio of network traffic samples that are incorrectly classified to the total network traffic samples. It is measured using the Eq (16).

$$ER = \sum_{i=1}^{n} \frac{S_{IC}}{S_i} * 100 \tag{16}$$

From Eq (16), Error Rate '$ER$' is computed depending on the samples involved '$S_i$' and the incorrectly classified samples '$S_{IC}$'. It is calculated in percentage (%). Error rate comparison of proposed GM-FVRL versus existing IPro [1], MACCA2-RF&RF [2], and Novel hybrid DL model based on CNN [10] is shown in Table 5.

Table 5 given above depicts $ER$ for 5000 dissimilar network traffic samples. X-axis denotes the network traffic samples obtained from 500 to 5000. The y-axis denotes the error rate. As

**Table 5. Error rate comparison of proposed GM-FVRL versus existing IPro [1], MACCA2-RF&RF [2], and novel hybrid DL model based on CNN [10].**

| Network samples | Error rate (%) | | | |
|---|---|---|---|---|
| | GM-FVRL | IPro | MACCA2-RF&RF | Novel hybrid DL model based on CNN |
| 500 | 2 | 3 | 6 | 2.75 |
| 1000 | 2.65 | 3.85 | 6.15 | 3.15 |
| 1500 | 2.85 | 5 | 6.85 | 4.15 |
| 2000 | 3 | 5.15 | 7 | 4.85 |
| 2500 | 3.15 | 5.75 | 7.15 | 5.25 |
| 3000 | 3.65 | 6 | 8 | 5.45 |
| 3500 | 4 | 7 | 8.15 | 6.15 |
| 4000 | 4.85 | 7.45 | 9 | 7 |
| 4500 | 6 | 7.85 | 9.15 | 7.15 |
| 5000 | 6.75 | 8 | 10 | 7.45 |

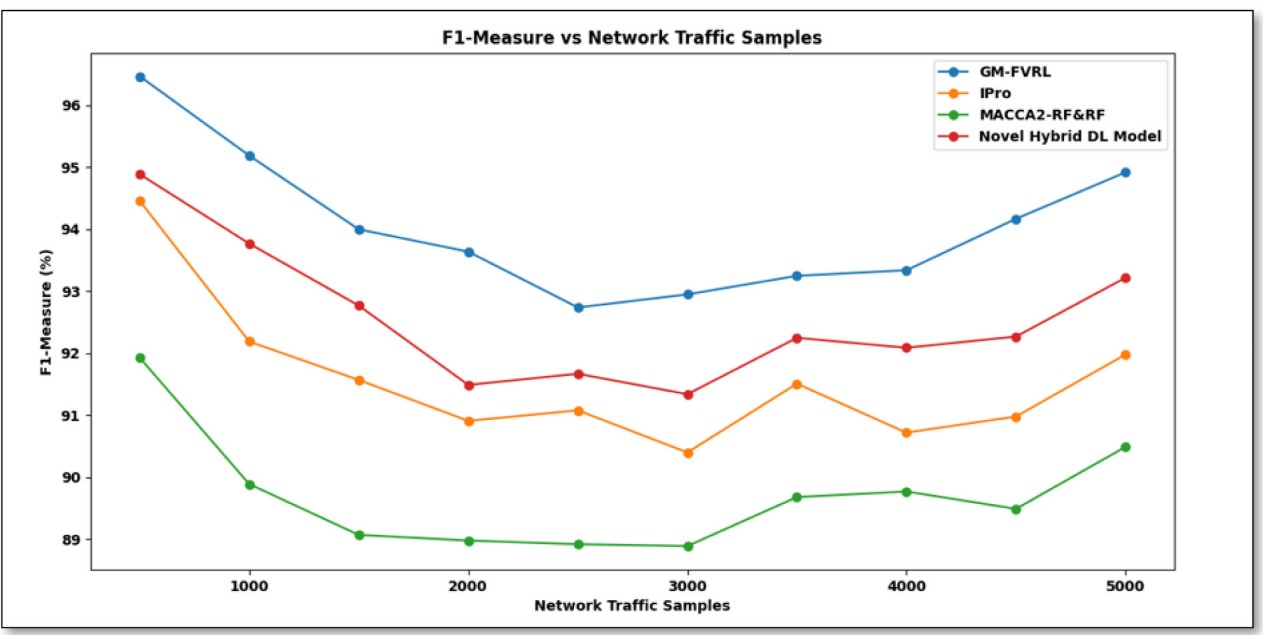

**Fig 6. F1-measure comparison of proposed GM-FVRL versus existing IPro [1], MACCA2-RF&RF [2], and novel hybrid DL model based on CNN [10].**

exposed in Table 4, GM-FVRL is better at achieving a minimum *ER* compared to existing methods. However, simulations were performed with 500 samples: 20 samples incorrectly classified using GM-FVRL, 30 samples using [1], 60 samples using [2], and 14 samples using [10]. From this outcome,*ER* was improved by the proposed GM-FVRL upon comparison with existing [1, 2, 10]. Reason for minimum *ER* is to use the Gauss Markov Correlation-based Network Traffic Feature Extraction method in GM-FVRL. Through using this method, relevant features are extracted with the help of the Gauss Markov function. After relevant features, classification was performed, therefore reducing the error rate. The GM-FVRL error rate is minimized by 35%, 51%, and 28% compared to [1, 2, 10].

**4.2.6. Case 6: F1-measure.** The F1 measure is measured as the average of *P* and *R* results. It is determined by the formula below,

$$F1 - \text{measure} = 2*\frac{Precision*Recall}{Precision + Recall}*100 \qquad (17)$$

From Eq (17), the F1-measure is computed depending on precision and recall results. It is calculated in percentage (%). Fig 6 provides an F1-measure comparison of proposed GM-FVRL versus existing IPro [1], MACCA2-RF&RF [2], and a Novel hybrid DL model based on CNN [10] varying the network traffic samples between 500 and 5000.

Fig 6 explains performance of the F1 measure. The experiment shows that the four methods increase the F1 measure according to the network traffic samples. However, the proposed GM-FVRL increases the F1 measure more than the other existing methods. F1-measure of existing [1, 2, 10] is achieved as 94.46, 91.93, and 94.89 while processing 500 network traffic samples. In addition, the proposed GM-FVRL increases the F1-measure to 96.47 with the same network traffic samples. From these results, it can be inferred that the F1-measure using the proposed GM-FVRL is comparatively better than existing [1, 2, 10]. This is due to applying

**Table 6. Comparison of all performance metrics for proposed GM-FVRL versus existing IPro [1], MACCA2-RF&RF [2], and novel hybrid DL model based on CNN [10].**

| Method | Proposed method | Existing method [1] | Existing method [2] | Existing method [10] |
|---|---|---|---|---|
| | GM-FVRL | IPro | MACCA2-RF&RF | Novel hybrid DL model based on CNN |
| Latency (ms) | 975.-01 ms | 1159.63 ms | 1433.63 ms | 1547.49 ms |
| Classification accuracy (%) | 96.11% | 94.09% | 92.25% | 94.66% |
| Precision (%) | 94.71% | 92.84% | 90.6% | 93.31% |
| Recall (%) | 93.46% | 90.40% | 88.87% | 91.87% |
| Error Rate (%) | 3.89% | 5.90% | 7.74% | 5.33% |
| F1-measure (%) | 94.06% | 91.58% | 89.71% | 92.57% |

the Network Traffic Feature Extraction model in proposed GM-FVRL for performing the relevant feature extraction steps. It provides network traffic and extracts mainly pertinent network traffic aspects depending on network environment changes. The comparison of average results indicates that the average of the F1-measure using the GM-FVRL is improved by 3%, 5%, and 2% when compared to [1, 2, 10].

The comparison of all performance metrics for proposed GM-FVRL versus existing IPro [1], MACCA2-RF&RF [2], and Novel hybrid DL model based on CNN [10] is shown in Table 6.

The proposed GM-FVRL method has performed much better than the existing [1, 2, 10] in latency, classification accuracy, $P$, $R$, error rate, and F1-measure by using SDN traffic dataset.

## 5. Conclusion

This paper has performed as evidence of abstraction while integrating ML through SDN, specifically for network traffic classification. It shows which network traffic categorization employing ML techniques bestows acceptable outcomes within an SDN environment. The network traffic categorization using the ML model is designed in SDN with an IoT framework. Network traffic classification is a traffic forecast important for network performance study. SDN on IoT bestows an exceptional platform for applying network traffic classification algorithms by GM-FVRL. The Gauss Markov Correlation-based Network, Traffic Feature Extraction method is employed to extort computationally efficient and relevant network traffic features for classification from SDN. Then, the Flow Balanced Radial-based ML model for network traffic categorization algorithm is established using extracted network traffic feature. Here, conditional checking for normal patterns and abnormality is measured based on the flow-balanced radial function. Therefore, it ensures precise network traffic classification. Simulation outcomes demonstrate which GM-FVRL method can classify network traffic behaviour tendency in an accurate and precise manner with minimum latency.

## Author Contributions

**Conceptualization:** Rajkumar Kulandaivel, Manikandan Ramachandran, Sathishkumar Veerappampalayam Easwaramoorthy.

**Data curation:** Sathishkumar Veerappampalayam Easwaramoorthy.

**Formal analysis:** Rajkumar Kulandaivel, Sathishkumar Veerappampalayam Easwaramoorthy.

**Funding acquisition:** Sathishkumar Veerappampalayam Easwaramoorthy, Jaehyuk Cho.

**Investigation:** Manikandan Ramachandran, Sathishkumar Veerappampalayam Easwaramoorthy.

**Methodology:** Sathishkumar Veerappampalayam Easwaramoorthy.

**Project administration:** Sathishkumar Veerappampalayam Easwaramoorthy.

**Resources:** Sathishkumar Veerappampalayam Easwaramoorthy.

**Software:** Rajkumar Kulandaivel, Sathishkumar Veerappampalayam Easwaramoorthy.

**Supervision:** Sathishkumar Veerappampalayam Easwaramoorthy, Jaehyuk Cho.

**Validation:** Sathishkumar Veerappampalayam Easwaramoorthy, Jaehyuk Cho.

**Visualization:** Sathishkumar Veerappampalayam Easwaramoorthy.

**Writing – original draft:** Rajkumar Kulandaivel, Manikandan Ramachandran, Sathishkumar Veerappampalayam Easwaramoorthy.

**Writing – review & editing:** Sathishkumar Veerappampalayam Easwaramoorthy.

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
