## [Decision Letter · Decision Letter 0]

5 Apr 2024

PONE-D-24-05742Gauss Markov and Flow Balanced Vector Radial Learning Network Traffic Classification on IoT with SDNPLOS ONE

Dear Dr. Cho,

Thank you for submitting your manuscript to PLOS ONE. After careful consideration, we feel that it has merit but does not fully meet PLOS ONE’s publication criteria as it currently stands. Therefore, we invite you to submit a revised version of the manuscript that addresses the points raised during the review process.

We look forward to receiving your revised manuscript.

Kind regards,

Muhammad Nasir Khan, PhD

Academic Editor

PLOS ONE

Journal Requirements:

"This work was supported the Korea Environmental Industry & Technology Institute (KEITI), with a grant funded by the Korea government, Ministry of Environment (The development of IoT-based technology for collecting and managing big data on environmental hazards and health effects), under Grant RE202101551 and partially supported by the Institute of Information and Communications Technology Planning and Evaluation (IITP) funded by the Korean Government, Ministry of Science and ICT (MSIT) (Implementation of Verification Platform for ICT Based Environmental Monitoring Sensor), under Grant 2019-0-00135"

"This work was supported by the Korea Environmental Industry & Technology Institute (KEITI), with a grant funded by the Korean government, Ministry of Environment (The development of IoT-based technology for collecting and managing big data on environmental hazards and health effects), under Grant RE202101551 and partially supported by the Institute of Information and Communications Technology Planning and Evaluation (IITP) funded by the Korean Government, Ministry of Science and ICT (MSIT) (Implementation of Verification Platform for ICT Based Environmental Monitoring Sensor), under Grant 2019-0-00135"

"This work was supported the Korea Environmental Industry & Technology Institute (KEITI), with a grant funded by the Korea government, Ministry of Environment (The development of IoT-based technology for collecting and managing big data on environmental hazards and health effects), under Grant RE202101551 and partially supported by the Institute of Information and Communications Technology Planning and Evaluation (IITP) funded by the Korean Government, Ministry of Science and ICT (MSIT) (Implementation of Verification Platform for ICT Based Environmental Monitoring Sensor), under Grant 2019-0-00135"

5. In the online submission form, you indicated that [The datasets used and/or analysed during the current study available from the corresponding author on reasonable request.]. 

Reviewers' comments:

Reviewer's Responses to Questions

**Comments to the Author**

1. Is the manuscript technically sound, and do the data support the conclusions?

Reviewer #1: Partly

Reviewer #2: Yes

2. Has the statistical analysis been performed appropriately and rigorously? 

Reviewer #1: N/A

Reviewer #2: Yes

3. Have the authors made all data underlying the findings in their manuscript fully available?

Reviewer #1: Yes

Reviewer #2: Yes

4. Is the manuscript presented in an intelligible fashion and written in standard English?

Reviewer #1: Yes

Reviewer #2: Yes

5. Review Comments to the Author

Reviewer #1: Before the Editor makes a decision, I suggest that the authors must take into account the following corrections:

1. I don't think the abbreviations used in the title are inspired.

2. I think the title needs to be reformulated to become more “friendly”.

In its current form, it doesn't really make meaning.

3. The "Introduction" section should be more concise. Also, Section 2.

4. What is the motivation of the values in matrix (1)?

5. Author must argue how the relation (2) was obtained, or indicate a bibliographic reference for its original form.

6. It is not clear how were obtained the relation (3).

7. The right member of relations (3) and (4) is the same. Then why the need for Cor[A_{FF}]?

8. Details on obtaining relation (8) are required.

9. Relation (11) is carelessly edited.

10. Details on obtaining the data in Tables are required.

11. From where were taken the data used in the graphic representations?

12. Some editing "glitches" need to be corrected.

13. Punctuations are used randomly. Insert comma or full stop after each and every equation accordingly.

14. I think the authors need to emphasize more clearly the contribution of the manuscript from a scientific point of view.

15. References are not uniformly written. In some references the volume, or the issue, or the pages are not specified.

16. Also, I think, the authors must strengthen the References section with some articles that use some similar techniques, to make the techniques used more plausible, for instance: The Effects of Fractional Time Derivatives in Porothermoelastic Materials Using Finite Element Method, Mathematics, 9(14) (2021), Art. No. 1606; On mixed problem in thermos-elasticity of type III for Cosserat media, Journal of Taibah University for Science, 16(1) (2022), 1264–1274.

If the authors take into account all these corrections, then this manuscript deserves to be published.

Reviewer #2: 1. In abstract, The existing approach helps to reduce accuracy. Is it meaningful the word "helps"?

2. Justify the need of two different parameters duration_sec and duration_nsec as in Table 2.

3. What is the impact for increasing the number of traffic samples to compute precision as in Table 4? There is no linear growth.

4. In figure 2, there is some hidden content in "features"

5. No clarity in Figure 2, no content in ellipse symbol and no proper interconnection

6. Highlight the causes of occurring network traffic.

7. Overall organization of the paper is quite novelty and noteworthy.

6. PLOS authors have the option to publish the peer review history of their article (what does this mean?). If published, this will include your full peer review and any attached files.

Reviewer #1: No

Reviewer #2: **Yes: **MARIKKANNAN M

---

## [Author Response · Author response to Decision Letter 0]

22 May 2024

Comment wise response is added as a separate file.

---

## [Decision Letter · Decision Letter 1]

17 Jul 2024

Gauss Markov and Flow Balanced Vector Radial Learning Traffic Classification in Software-defined Networking

PONE-D-24-05742R1

Dear Dr. Cho,

We’re pleased to inform you that your manuscript has been judged scientifically suitable for publication and will be formally accepted for publication once it meets all outstanding technical requirements.

Kind regards,

Muhammad Nasir Khan, PhD

Academic Editor

PLOS ONE

Additional Editor Comments (optional):

Reviewers' comments:

Reviewer's Responses to Questions

**Comments to the Author**

1. If the authors have adequately addressed your comments raised in a previous round of review and you feel that this manuscript is now acceptable for publication, you may indicate that here to bypass the “Comments to the Author” section, enter your conflict of interest statement in the “Confidential to Editor” section, and submit your "Accept" recommendation.

Reviewer #2: All comments have been addressed

Reviewer #3: All comments have been addressed

2. Is the manuscript technically sound, and do the data support the conclusions?

Reviewer #2: Yes

Reviewer #3: Yes

3. Has the statistical analysis been performed appropriately and rigorously? 

Reviewer #2: No

Reviewer #3: Yes

4. Have the authors made all data underlying the findings in their manuscript fully available?

Reviewer #2: Yes

Reviewer #3: Yes

5. Is the manuscript presented in an intelligible fashion and written in standard English?

Reviewer #2: Yes

Reviewer #3: Yes

6. Review Comments to the Author

Reviewer #2: In my earlier question no 1, Reduction of accuracy in any approach is feasible?

Question no 2: Need of two different parameters seconds and nanoseconds (could be considered either seconds or nanoseconds)

Reviewer #3: The revised version of the article has significantly improved, addressing the major concerns and suggestions from the initial review. The methodology is robust, data analysis is comprehensive, and the discussion provides valuable insights. The manuscript is well-structured and clearly written, demonstrating high scholarly rigor. I commend the authors for their diligent revisions and believe this work will be a valuable addition to the literature.

7. PLOS authors have the option to publish the peer review history of their article (what does this mean?). If published, this will include your full peer review and any attached files.

Reviewer #2: **Yes: **MARIKKANNAN M

Reviewer #3: **Yes: **Fawad Naseer

---

## [Editor Report · Acceptance letter]

20 Sep 2024

PONE-D-24-05742R1 

PLOS ONE

Dear Dr. Cho, 

I'm pleased to inform you that your manuscript has been deemed suitable for publication in PLOS ONE. Congratulations! Your manuscript is now being handed over to our production team.

Kind regards, 

on behalf of

Dr. Muhammad Nasir Khan 

%CORR_ED_EDITOR_ROLE%

PLOS ONE